# Preclinical systematic review of CCR5 antagonists as cerebroprotective and stroke recovery enhancing agents

Ayni Sharif[1,2†], Matthew S Jeffers[1,2†], Dean A Fergusson[1,2,3], Raj Bapuji[1], Stuart G Nicholls[4], John Humphrey[5], Warren Johnston[5], Ed Mitchell[5], Mary-Ann Speirs[5], Laura Stronghill[5], Michele Vuckovic[5], Susan Wulf[5], Risa Shorr[6], Dar Dowlatshahi[2,7,8], Dale Corbett[8], Manoj M Lalu[1,2,8,9]*

[1]Clinical Epidemiology Program, Blueprint Translational Research Group, Ottawa Hospital Research Institute, Ottawa, Canada; [2]School of Epidemiology and Public Health, University of Ottawa, Ottawa, Canada; [3]Department of Medicine, University of Ottawa, Ottawa, Canada; [4]Office for Patient Engagement in Research Activity (OPERA), Ottawa Hospital Research Institute, Ottawa, Canada; [5]Patient Partner, Blueprint Translational Research Group, Ottawa Hospital Research Institute, Ottawa, Canada; [6]Ottawa Hospital Library Services, The Ottawa Hospital, Ottawa, Canada; [7]Division of Neurology, Department of Medicine, The Ottawa Hospital, Ottawa, Canada; [8]Department of Cellular and Molecular Medicine, Faculty of Medicine, University of Ottawa, Ottawa, Canada; [9]Department of Anaesthesiology and Pain Medicine, The Ottawa Hospital, Ottawa, Canada

*For correspondence: mlalu@toh.ca

[†]These authors contributed equally to this work

Competing interest: The authors declare that no competing interests exist.

## eLife Assessment

This study is **important**, and the findings add substantially to the evidence base regarding CCR5 antagonist drugs for neuroprotection and stroke management. The authors adhered to the expected systematic review and meta-analysis standards, and the presented evidence is **convincing**.

**Abstract** C-C chemokine receptor type 5 (CCR5) antagonists may improve both acute stroke outcome and long-term recovery. Despite their evaluation in ongoing clinical trials, gaps remain in the evidence supporting their use. With a panel of patients with lived experiences of stroke, we performed a systematic review of animal models of stroke that administered a CCR5 antagonist and assessed infarct size or behavioural outcomes. MEDLINE, Web of Science, and Embase were searched. Article screening and data extraction were completed in duplicate. We pooled outcomes using random effects meta-analyses. We assessed risk of bias using the Systematic Review Centre for Laboratory Animal Experimentation (SYRCLE) tool and alignment with the Stroke Treatment Academic Industry Roundtable (STAIR) and Stroke Recovery and Rehabilitation Roundtable (SRRR) recommendations. Five studies representing 10 experiments were included. CCR5 antagonists reduced infarct volume (standard mean difference −1.02; 95% confidence interval −1.58 to −0.46) when compared to stroke-only controls. Varied timing of CCR5 administration (pre- or post-stroke induction) produced similar benefit. CCR5 antagonists significantly improved 11 of 16 behavioural outcomes reported. High risk of bias was present in all studies and critical knowledge gaps in the preclinical evidence were identified using STAIR/SRRR. CCR5 antagonists demonstrate promise; however, rigorously designed preclinical studies that better align with STAIR/SRRR recommendations and downstream clinical trials are warranted. Prospective Register of Systematic Reviews (PROSPERO CRD42023393438).

**eLife digest** Most promising new treatments for stroke patients face a significant challenge. Preclinical laboratory experiments on animal models may show potential, however thousands are rejected in the expensive and complicated process of human clinical trials. How should researchers design preclinical experiments to increase the chances of success in later human studies? One possible way is to match the type of evidence used by preclinical researchers with the expectations of clinicians and patients. Improving communication between these groups could ensure that preclinical studies contain experiments that are more relevant to a clinical trial setting.

With this in mind, Sharif, Jeffers et al. developed criteria for preclinical studies by analyzing multiple studies and collaborating with patient partners with lived experiences of strokes. This allowed the researchers to assess the readiness of a promising class of drugs known as CCR5 antagonists (which are already approved for medical conditions other than stroke) for clinical trials of stroke treatment.

Specifically, Sharif, Jeffers et al. screened for studies that included preclinical experiments in rats and mice that investigated the effect of CCR5 antagonists on brain recovery after stroke. From these, ten experiments which varied in the drug used, dose, and timing of treatment were compared, showing that in all cases the total area of the brain affected by stroke was reduced by the treatment. The CCR5 antagonists were also effective in treatments applied before or after a stroke event. Throughout the study the researchers worked with patient partners to assess the treatment outcomes that align with patient priorities, concluding that this factor should play a more significant role in the design of preclinical studies.

Overall, the findings show that CCR5 antagonists could represent a promising treatment for stroke patients and that future preclinical studies should involve patients and clinicians in experimental design to increase the chances of success in human trials. The proposed approach could be used in future studies when assessing which therapies should progress to human trials.

## Introduction

C-C chemokine receptor type 5 (CCR5) is expressed across a variety of leukocyte subtypes, endothelial cells, and cell types in the brain (e.g., neurons, microglia, astrocytes), and is thought to play crucial roles in post-stroke neuroinflammation, blood–brain barrier repair, and neuronal survival/repair processes (*Jing et al., 2023*). CCR5 antagonists have emerged as potential therapeutic candidates for stroke, demonstrating both cerebroprotection and improved neural repair/recovery in preclinical animal models (*Li et al., 2016*; *Takami et al., 2002*; *Chen et al., 2022*; *Yan et al., 2021*; *Joy et al., 2019*). However, no CCR5 antagonist drug has an approved indication in the stroke context, necessitating studies to establish safety and efficacy this population. This has led to an ongoing clinical trial to investigate efficacy of CCR5 antagonists in combination with post-stroke rehabilitation (*Dukelow, 2023*). Assessment of the preclinical evidence supporting CCR5's role in stroke is needed to identify areas of potential benefit, and knowledge gaps, that should be addressed by future preclinical research (*O'Collins et al., 2006*).

The stroke cerebroprotection and recovery communities have advocated for alignment of preclinical and clinical study parameters through publication of consensus recommendations for preclinical research, in an effort to enhance the translation of new stroke therapies (*Corbett et al., 2017*; *Bosetti et al., 2017*; *Lyden et al., 2021*). Examples include identification of more sensitive and clinically relevant preclinical outcome measures and incorporation of potentially important effect modifiers of treatment efficacy, such as age, sex, and stroke-related comorbidities (hypertension, diabetes, etc.) (*Corbett et al., 2017*; *Bosetti et al., 2017*; *Lyden et al., 2021*). These recommendations aim to improve the translation of novel stroke therapeutics from preclinical to clinical populations, but the degree to which preclinical evidence for CCR5 antagonists satisfy these recommendations is unknown.

We sought to comprehensively evaluate the preclinical evidence for CCR5 antagonist drugs as both cerebroprotective and stroke recovery-promoting agents (*Corbett et al., 2017*; *Lyden et al., 2021*). Both perspectives are necessary to fully understand the therapeutic potential of stroke-related treatments, as distinct biological principles, time windows for treatment, and outcomes of interest underpin each of these treatment domains (*Carmichael, 2016*; *Murphy and Corbett, 2009*). We conducted a systematic review and meta-analysis of the CCR5 literature in conjunction with a panel of

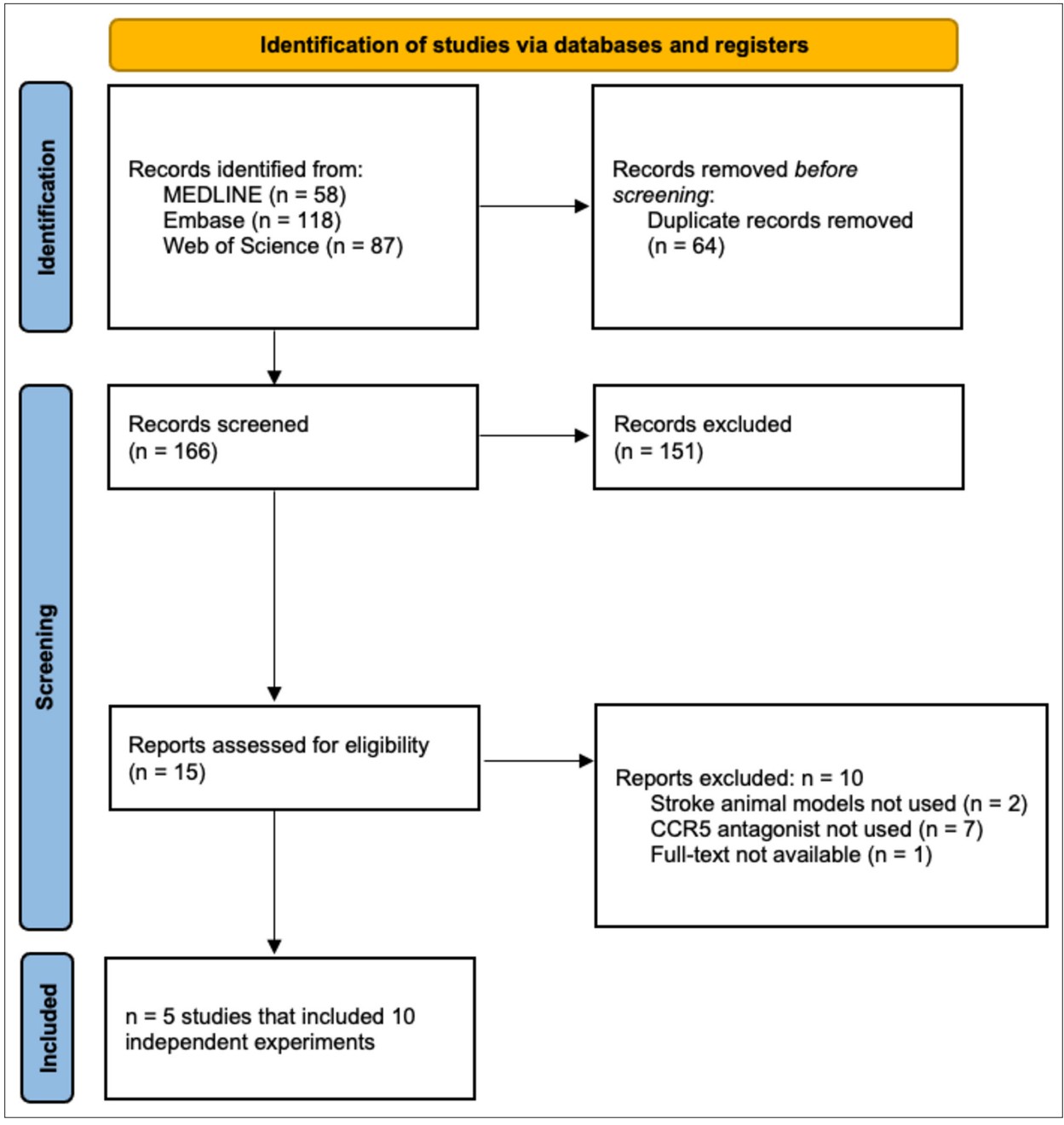

**Figure 1.** Preferred Reporting Items for Systematic Reviews and Meta-Analysis (PRISMA) flow diagram.

individuals with lived experiences of stroke (patient partners). This review sought to explore how the preclinical evidence for CCR5 antagonist drugs aligned with guidance for preclinical stroke research provided by previous expert committees and the parameters of an ongoing clinical trial (The Canadian Maraviroc Randomized Controlled Trial To Augment Rehabilitation Outcomes After Stroke, CAMAROS, NCT04789616) (*Corbett et al., 2017*; *Bosetti et al., 2017*; *Lyden et al., 2021*).

## Results
### Study selection
Our search identified 263 citations, which was reduced to 166 unique studies after deduplication. Five studies representing 10 experiments met the eligibility criteria (*Figure 1*; *Li et al., 2016*; *Takami et al.,*

**Table 1.** Summary of study and animal model characteristics of included articles.

| Author | Year | Country | Species | Strain | Stroke type | Stroke model | Sex | Weight | Age |
|---|---|---|---|---|---|---|---|---|---|
| *Li et al., 2016* | 2016 | China | Rat | Wistar | Ischaemic | Intraluminal suture | Male | 260–300 g | N/A |
| *Takami et al., 2002* | 2002 | Japan | Mice | ddY | Ischaemic | Intraluminal suture | Male | N/A | 4 weeks |
| *Chen et al., 2022* | 2022 | China | Mice | C57/BL6 | Ischaemic | Permanent middle cerebral artery occlusion | Male | 25–30 g | 8–10 weeks |
| *Yan et al., 2021* | 2021 | China | Mice | CD1 | Haemorrhagic | Intracerebral haemorrhagic | Male | 30–40 g | N/A |
| *Joy et al., 2019* | 2019 | USA | Mice | C57/BL6 | Ischaemic | Photothrombotic | Male | 25–30 g | 8–20 weeks |

*2002*; *Chen et al., 2022*; *Yan et al., 2021*; *Joy et al., 2019*). Herein, 'studies' refer to the published articles as a unit, while 'experiments' refer to distinct investigations within each published article used to test various hypotheses (i.e., a subunit of 'studies' comprised of a select cohort of animals).

## Study and animal model characteristics

Most studies used ischaemic stroke ($n$ = 4/5). This was induced via intraluminal suture ($n$ = 2), cauterization ($n$ = 1), or photothrombosis techniques ($n$ = 1; *Table 1*). Haemorrhagic stroke was induced in one study via autologous whole blood injection. All studies used mouse ($n$ = 4) or rat ($n$ = 1) models, comprised of exclusively male animals ($n$ = 5). Relevant stroke comorbidities highlighted by patient partners and Stroke Treatment Academic Industry Roundtable/Stroke Recovery and Rehabilitation Roundtable (STAIR/SRRR) recommendations (e.g., hypertension, diabetes) were not used in any study.

## Intervention characteristics

Maraviroc was used in six of the experiments ($n$ = 6/10), TAK-779 in three of the experiments ($n$ = 3), and D-Ala-Peptide T-Amide (DAPTA) in one ($n$ = 1; *Table 2*). These CCR5 antagonists were delivered intraperitoneally ($n$ = 4), intranasally ($n$ = 2), intracerebroventricularly ($n$ = 2), subcutaneously ($n$ = 1), and intravenously ($n$ = 1) at a dose range from 0.01 to 100 mg/kg. Most studies delivered a single dose of the drug ($n$ = 6); experiments with multiple administrations ($n$ = 4) ranged from 3 to 63 doses. Time of initial treatment administration varied widely. Two studies (three experiments) administered treatment pre-stroke (10–15 min) (*Li et al., 2016*; *Takami et al., 2002*) and four studies (six experiments) in the acute, potentially cerebroprotective, post-stroke period (50 min to 24 hr post-stroke) (*Takami et al., 2002*; *Chen et al., 2022*; *Yan et al., 2021*; *Joy et al., 2019*). One study (one experiment) was conducted in the late subacute/early chronic period beginning at 3–4 weeks post-stroke, which would be oriented towards recovery, rather than cerebroprotective, effects (*Joy et al., 2019*). Patient partners had identified several a priori interests (physical therapy alongside CCR5 administration, spasticity), which were also aligned with SRRR recommended considerations for preclinical stroke recovery studies (*Corbett et al., 2017*). These were not reported in any included studies. We also extracted a list of outcomes used to determine CCR5's potential mechanisms of action (*Appendix 1— figure 1*, *Appendix 1—table 1*).

## Meta-analysis of infarct volume

Infarct volume was reported in six experiments ($n$ = 6/10) from four different studies with an overall pooled analysis demonstrating marked cerebroprotection with CCR5 antagonists (standardized mean difference [SMD] –1.02, 95% confidence interval [CI] –1.58 to –0.46, p < 0.0001, $I^2$ = 34%; *Figure 2*). Five of these experiments measured infarct volume at 1–3 days post-stroke, and one experiment measured infarct volume at a delayed time point of 63 days post-stroke. No significant differences between pre- or post-stroke administration were observed (p = 0.47). Post hoc sensitivity analysis removing one experiment with extreme values *Li et al., 2016* demonstrated that cerebroprotection in the remaining two experiments remained statistically significant while reducing heterogeneity (SMD –0.81, 95% CI –1.25 to –0.37, p < 0.001, $I^2$ = 0%). A second sensitivity analysis excluded one study that measured infarct volume in mm (*Takami et al., 2002*) so that all other studies could be meta-analysed using mean differences on the percentage scale. This demonstrated a similar cerebroprotective effect as the other analyses (*Figure 2—figure supplement 1*; MD –9.1%, 95% CI –11.6 to

**Table 2.** Summary of intervention characteristics.

| Author | Drug | Dose (mg/kg) | Route | Timing | Doses (total #) | Outcomes measured (treatment n/control n) | Outcome window (post-stroke) |
|---|---|---|---|---|---|---|---|
| *Li et al., 2016* | DAPTA (D-Ala-Peptide T-Amide) | 0.01 | SC | 15-min pre-stroke | 1 | • Infarct volume (5/5)<br>• Neurological deficit score (5/5) | 24 hr |
| *Takami et al., 2002* | TAK-779 | 5 | ICV | 10-min pre-stroke | 1 | • Infarct volume (10/6) | 48 hr |
| | | 50 | ICV | 10-min pre-stroke | | • Infarct volume (13/6) | |
| | | 0.25 | IV | 50-min post-stroke | | • Infarct volume (17/18) | |
| *Chen et al., 2022* | Maraviroc | 20 | IP | 1.5-, 24-, and 48-hr post-stroke | 3 | • Infarct volume (5/5)<br>• Longa score (5/5)<br>• Neurological deficit score (5/5) | 72 hr |
| *Yan et al., 2021* | Maraviroc | 0.15 | IN | 1-hr post-stroke | 1 | • Garcia test (6/6)<br>• Limb placement (6/6)<br>• Corner turn test (6/6) | 72 hr |
| | | | | | | • Foot fault (8/8)<br>• Rotarod (8/8) | 3 weeks |
| | | | | | | • Probe quadrant duration (8/8) | 25 days |
| | | | | 24-hr post-stroke | | • Garcia test (6/6)<br>• Limb placement (6/6)<br>• Corner turn test (6/6) | 72 hr |
| *Joy et al., 2019* | Maraviroc | 100 | IP | 24-hr post-stroke through daily injections for 9 weeks | 63 | • Infarct volume (5/5) | 9 weeks |
| | | | | | | • Grid walk (10/10)<br>• Forelimb (10/10) | 8 weeks |
| | | | | 24-hr post-stroke then daily for 3 weeks | 21 | • Grid walk (10/10)<br>• Cylinder test (9/8) | 9 weeks |
| | | | | 3–4 weeks post-stroke then daily for 11 weeks | 56 | • Grid walk (9/9)<br>• Cylinder test (9/9) | 11 weeks |

ICV = intracerebroventricular; IN = intranasal; IP = intraperitoneal; IV = intravenous; SC = subcutaneous.

–6.7%, p < 0.001, $I^2$ = 0%). Further subgroup analyses by route of administration, time of administration, stroke model, species, CCR5 antagonist, dose, and whether behaviour tests were assessed are described in *Figure 2—figure supplement 2*, *Figure 2—figure supplement 3*, and *Figure 2—figure supplement 4*.

## Synthesis of behavioural outcomes without meta-analysis

Motor behavioural outcomes were reported in six experiments from three studies and represented seven different behavioural tasks. An additional study reported motor behavioural outcomes without standard deviations or standard errors, and thus could not be included (authors did not respond to email requests for data) (*Chen et al., 2022*). A cognitive outcome (Morris Water Maze) was measured in one study. Overall, CCR5 inhibition was effective in 11 of 16 behavioural outcomes tested (*Figure 3*). Meta-analysis and planned subgroup analysis were were not possible due to an inadequate number of studies for each given outcome measure, which necessitated the synthesis without meta-analysis presented below (*Campbell et al., 2020*).

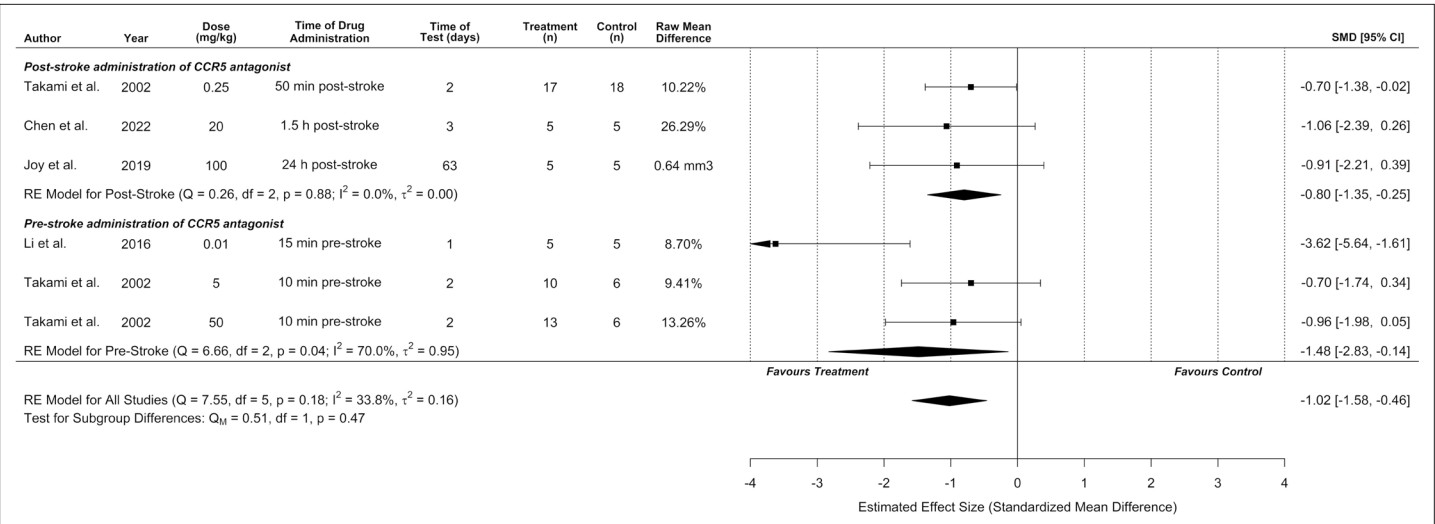

**Figure 2.** C-C chemokine receptor type 5 (CCR5) antagonists reduce infarct volume. Data is presented as a forest plot with standardized mean differences and 95% confidence intervals. Effect sizes <0 favours drug treatment and >0 favours control. Data is stratified by timing of CCR5 antagonist administration (pre- or post-stroke induction). The 'RE Model for All Studies' represents a pooled estimate of the CCR5 antagonist drug effect on infarct volume from all studies combined. Separate pooled estimates are also reported for post- and pre-stroke CCR5.

The online version of this article includes the following figure supplement(s) for figure 2:

**Figure supplement 1.** Sensitivity analysis for all included studies that reported infarct volume on a percentage scale.

**Figure supplement 2.** Subgroup analysis for all included studies of pre- and post-stroke C-C chemokine receptor type 5 (CCR5) antagonist administration that reported infarct volume.

**Figure supplement 3.** Subgroup analysis for all included studies of post-stroke drug administration of a C-C chemokine receptor type 5 (CCR5) antagonist that reported infarct volume.

**Figure supplement 4.** Subgroup analysis for all included studies of pre-stroke drug administration of a C-C chemokine receptor type 5 (CCR5) antagonist that reported infarct volume.

Behavioural outcomes are presented by time of CCR5 antagonist administration, as discussed in the Intervention Characteristics section, as administration time directly relates to the treatment context and patient population to which the results apply (i.e., acute cerebroprotection vs. late subacute/early chronic neural repair). Li et al. reported that pre-stroke administration (relevant to surgical contexts with high risk of thrombosis) of DAPTA did not result in significantly greater performance on the neurological deficit score (*Li et al., 2016*).

Regarding acute post-stroke administration times with the potential for *cerebroprotective* effects (up to 24-hr post-stroke based on observed infarct reductions in *Figure 2*), Yan et al. observed that CCR5 antagonist (maraviroc) administration 1-hr following stroke resulted in greater motor performance on the corner test, limb placement, modified Garcia, foot fault, and rotarod tasks compared to vehicle-treated controls. Cognitive outcomes in the Morris Water Maze task (proportion of time spent in the probe quadrant) were also improved by this 1-hr post-stroke administration. Significantly improved motor performance was not observed if CCR5 antagonist was administered 24-hr post-stroke, using this model of *intracerebral haemorrhage* (*Figure 3*; *Yan et al., 2021*). In contrast, using a *focal ischaemia* model, Joy et al. observed that CCR5 antagonist (maraviroc) administration initiated 24-hr post-stroke and continued daily for 3 weeks (which could potentially represent a mix of cerebroprotective and recovery-induced effects) resulted in greater performance on the cylinder and grid walk tasks compared to vehicle-treated controls.

Joy et al. also performed one experiment with *late subacute/early chronic* administration (a potentially critical time for neural repair) initiated at 3–4 weeks post-stroke and continued daily for 11 weeks, that demonstrated significantly improved performance for the grid walk, but not cylinder, task (*Joy et al., 2019*). This experiment was initiated outside the plausible range for a cerebroprotective effect, implying behavioural improvement involved *recovery-promoting mechanisms*. However, equivalent infarct volumes were not demonstrated between the treated and control groups in this cohort, which could potentially lead to confounding effects.

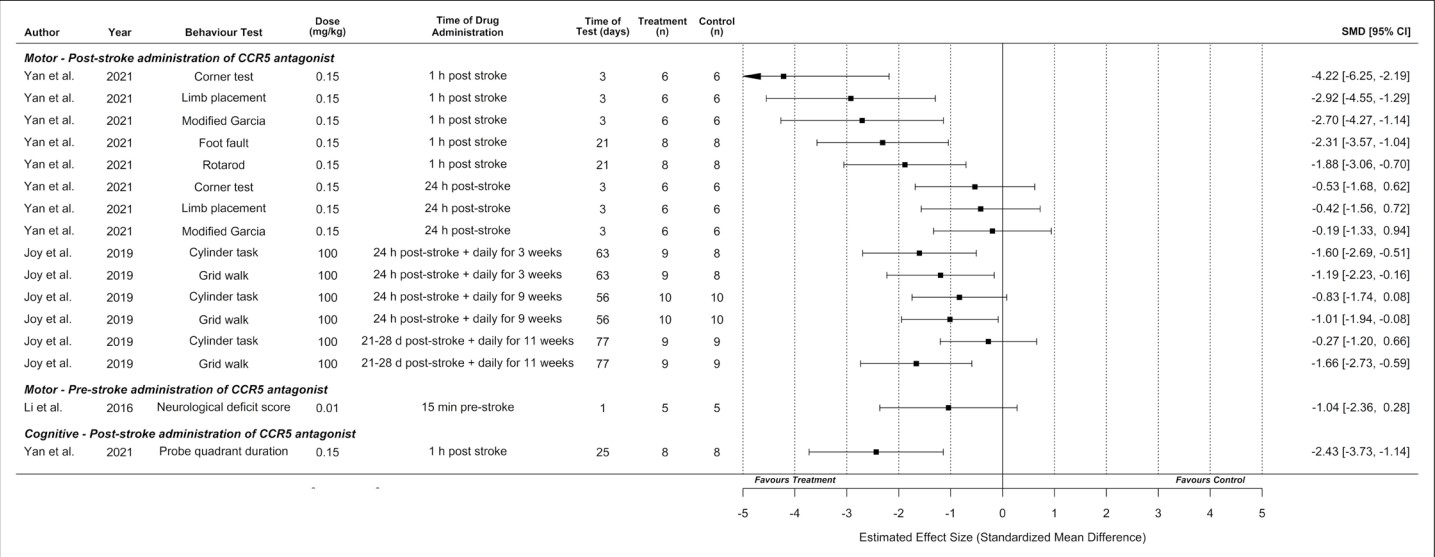

**Figure 3.** Synthesis without meta-analysis for all included preclinical C-C chemokine receptor type 5 (CCR5) antagonist studies that reported motor and/or cognitive behavioural outcomes. Data is presented as a forest plot with a standardized mean difference and 95% confidence intervals. Effect sizes <0 favours drug treatment and >0 favours control.

## Risk of bias

All articles had a 'high' risk of bias in at least one domain of the Systematic Review Centre for Laboratory Animal Experimentation (SYRCLE) tool (*Hooijmans et al., 2014*). Most domains within each study demonstrated an 'unclear' risk of bias. All studies reported randomizing animals; however, as commonly observed in the preclinical literature, (*Avey et al., 2016*; *Fergusson et al., 2019b*; *Fergusson et al., 2019a*) only one of these studies provided sufficient detail to ensure that the randomization method had a low risk of bias. The SYRCLE domains with the highest risk of bias were incomplete outcome data, with 80% of studies (*n* = 4/5) failing to provide complete data for all animals initially included in the study, as well as selective outcome reporting, with 60% (*n* = 3/5; all studies of maraviroc) not providing complete data for all expected outcomes discussed in the methods (*Figure 4*).

## Comprehensiveness of preclinical evidence and alignment with clinical trials

We assessed comprehensiveness of the preclinical evidence using the STAIR and SRRR recommendations (*Corbett et al., 2017*; *Lyden et al., 2021*; *Finklestein et al., 1999*; *Fisher et al., 2009*) as well as alignment with study parameters of the CAMAROS trial (*Dukelow, 2023*). A summary of assessment items from STAIR XI is provided in *Table 3*, with additional items from STAIR I, VI, and SRRR recommendations included in *Table 4*.

For CCR5 antagonists as a post-stroke cerebroprotectant, the overall body of evidence satisfied all five STAIR XI domains assessing 'candidate treatment qualification' (*Table 3*). Overall, a range of doses and clinically relevant administration times for cerebroprotection were evaluated across a variety of motor and cognitive behavioural domains. All studies tested both behavioural and histological outcomes and demonstrated cerebroprotective effects, but most studies failed to measure and control post-stroke temperature, which could potentially confound the observed cerebroprotection (*Table 4*; *Corbett and Nurse, 1998*). Most histological measurements were also assessed at <72 hr, which could confound the observed cerebroprotective effects if cell death was merely delayed (*Corbett and Nurse, 1998*). For CCR5 antagonists as a post-stroke recovery-inducing treatment, one experiment assessed the effects of initiating CCR5 administration in a similar post-stroke phase as the CAMAROS trial. This experiment (*Joy et al., 2019*) did not demonstrate that each treatment group had equivalent baseline stroke volumes, which may potentially confound observed behavioural effects. Furthermore, the maximum dose used in mice (100 mg/kg) was not sufficient to

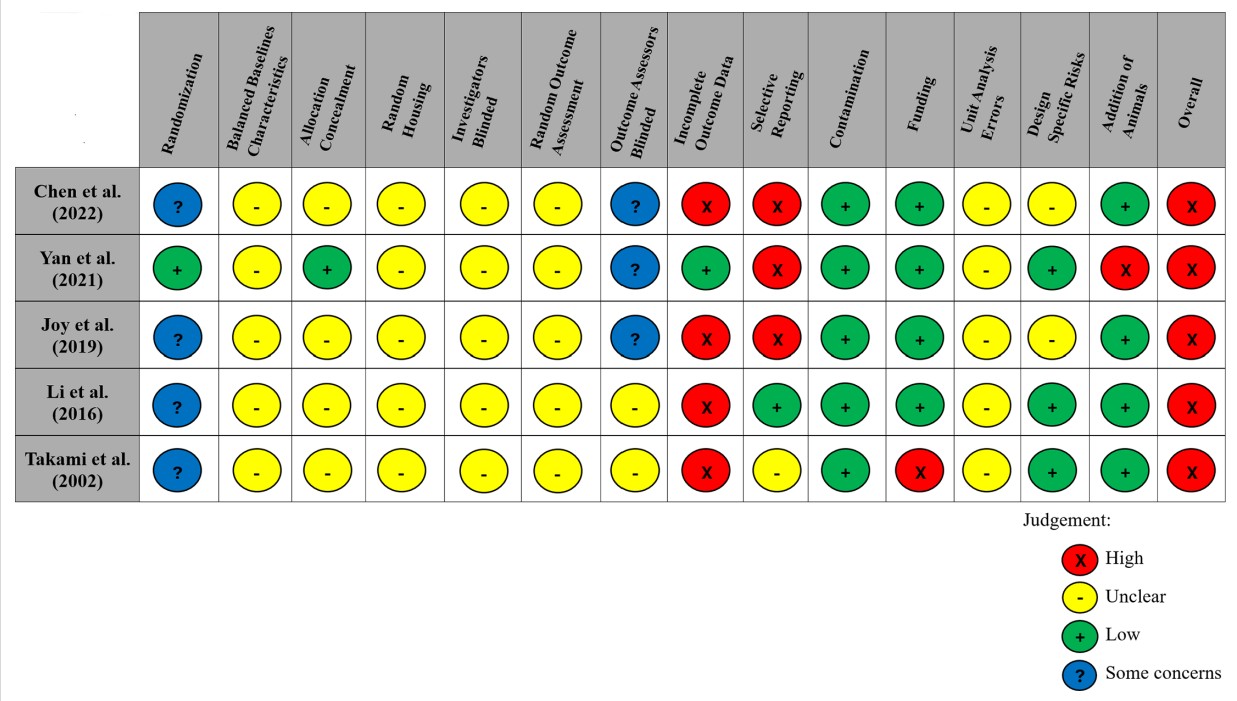

**Figure 4.** Modified risk of bias traffic light plot in accordance with the Systematic Review Centre for Laboratory Animal Experimentation (SYRCLE) tool. Yellow represents an unclear risk of bias, green represents a low risk of bias, and red represents a high risk of bias. Blue represents some concerns of a risk of bias. The risk of bias was 'unclear' across all studies for the domains of baseline characteristics because of missing data in the studies, random housing because no details on this domain were reported, and random outcome assessment because no details of how cohorts of animals were selected to perform certain outcomes nor how the order of outcome assessment proceeded. Four studies did not report on allocation concealment, and two studies did not report on blinding investigators and outcome assessors and were deemed as having an 'unclear' risk of bias. 80% of studies exhibited a 'high' risk for incomplete outcome data. Similarly, three studies (60%) had a 'high' risk of selective outcome reporting since all expected outcomes discussed in the methods of the articles did not align with their results. Other potential sources of bias considered included the source of funding (industry funded), contamination of pooling drugs (additional treatment which might influence or bias the result), unit error analysis (all animals receiving the same intervention are caged together, but analysis was conducted as if every single animal was one experimental unit), design-specific risks of bias (reporting details of which animals performed the same or different outcomes), and the addition of new animals to replace dropouts from the original population. Two studies (40%) had a 'high' risk in at least one of these additional categories.

attain cerebrospinal fluid levels of maraviroc observed in humans using the CAMAROS dosing regime (mice: 13.8 ± 5.4 ng/ml; humans 33.6–60 ng/ml) (*Joy et al., 2019*).

Areas of concern were identified in all STAIR XI domains assessing 'preclinical assessment and validation' (*Table 3*). Although adult animals were used in most of the preclinical studies, the reported weights and ages of these animals corresponded to young adults, rather than the aged adults that better represent the stroke population. All experiments used only male rodents that were free of common stroke comorbidities. It was also unclear if behavioural testing was performed during the inactive circadian phase or active (dark) phase, which could result in confounding if CCR5 antagonists affect arousal of animals during their inactive period (*MacLellan et al., 2011*). Furthermore, none of the studies had their protocols or results directly replicated by an independent laboratory, or across multiple sites in a multilaboratory study (*Corbett et al., 2017*; *Hunniford et al., 2023*). Regarding stroke recovery, no studies assessed behavioural effects on upper extremity skilled reaching/grasping or potential interactions of CCR5 antagonists with rehabilitative therapies or established recanalization procedures (*Table 4*; *Balbinot et al., 2018*; *Jeffers and Corbett, 2018*; *Jeffers et al., 2018*; *Wechsler et al., 2023*). These elements are highly relevant to the CAMAROS trial, as one of the primary outcomes of this trial is upper extremity performance on the Fugl-Meyer and maraviroc administration will be paired with an 8-week exercise program. These findings were supported by the PRIMED (*Li et al., 2016*) tool, which resulted in a Readiness for Translation Score of 'medium' (on a scale of 'low', 'medium', and 'high'). This tool highlighted similar promising elements, as well as weaknesses, as our analysis above (*Tables 3 and 4*), identifying the limited preclinical evidence

**Table 3.** Alignment of included preclinical studies with STAIR recommendations and CAMAROS trial parameters.

| Recommendation | Overall preclinical evidence | CAMAROS trial alignment | Notes |
|---|---|---|---|
| **Candidate treatment qualification** | | | |
| Establish treatment dose–response | Yes | No | <ul><li>**Preclinical** – doses across all studies from 0.01 to 100 mg/kg</li><li>Six doses tested for infarct volume, three doses for behavioural effects</li><li>Joy et al. demonstrated that 100 mg/kg of maraviroc in mice results in cerebrospinal fluid levels lower than in humans (13.8 ± 5.4 vs. 33.6–60 ng/ml)</li><li>**CAMAROS** – participants take 300 mg doses twice daily for 8 weeks</li><li>**Interpretation** – maximum preclinical dose may not represent clinical levels</li></ul> |
| Treatment given after clinically relevant delayed times (1- to 4.5-hr post-stroke) | Yes | Yes | <ul><li>**Preclinical** – 3/5 studies (all maraviroc) administered drug from 1 to 24 hr post-stroke (acute phase)</li><li>Joy et al. initiated maraviroc administration at 20 days post-stroke for 11 weeks (subacute phase)</li><li>**CAMAROS** – participants recruited between 5 days and 6 weeks post-stroke (acute/early subacute)</li><li>**Interpretation** – one experiment in a time window relevant to clinical trial</li></ul> |
| Both histologic and behavioural outcomes at acute (1–3 days) and chronic (7–30 days) time points | Yes | Yes | <ul><li>**Preclinical** – 2/5 studies included both histologic and behavioural outcomes</li><li>Mainly tasks of spontaneous movement (e.g., cylinder) or motor coordination (e.g., grid walk)</li><li>Tests ranged from 3 days (acute) to 11 weeks (chronic) post-stroke for all included studies, but short-term assessment may only delay cell death (*Corbett and Nurse, 1998*)</li><li>**CAMAROS** – motor learning (Fugl-Meyer Upper Extremity Assessment Score and 10-Meter Walk Test Score) measured as the primary outcome at baseline, 4 week (late subacute), 8 week (late subacute), and 6 month (chronic)</li><li>**Interpretation** – challenging to directly compare preclinical and clinical motor tasks, but testing range is comparable. Most relevant Joy et al. experiment (subacute administration) did not measure histologic outcomes</li></ul> |
| Treatment reaches presumed target and causes expected physiological effects that can be assessed with a clinically relevant biomarker | Yes | NA | <ul><li>**Preclinical** – 5/5 studies demonstrated trends for CCR5 antagonism to reduce infarct volume. Plausible that CCR5 influences stroke-relevant mechanisms</li><li>**CAMAROS** – does not include outcomes to assess treatment mechanisms</li></ul> |
| Treatment able to pass the blood–brain barrier | Yes | NA | <ul><li>**Preclinical** – 1/5 studies used mass spectrometry to demonstrate that maraviroc is present in the brain and cerebrospinal fluid</li><li>**CAMAROS** – does not include outcomes to assess presence of drug in the brain</li></ul> |
| **Preclinical assessment/validation** | | | |
| Aging/adult age | No | No | <ul><li>**Preclinical** – 4/5 studies used rodents with weights/ages consistent with young adulthood; 0/5 studies used rodents ≥10 months old.</li><li>**CAMAROS** – age ≥18 year old adult participants eligible for recruitment</li><li>**Interpretation** – preclinical studies consistent with CAMAROS eligibility criteria; however, no preclinical study examined middle-aged or elderly animals that would be more representative of the trial and overall stroke populations</li></ul> |

*Table 3 continued on next page*

*Table 3 continued*

| Recommendation | Overall preclinical evidence | CAMAROS trial alignment | Notes |
|---|---|---|---|
| Male and female animals | No | No | • **Preclinical** – 0/5 studies included female animals<br>• **CAMAROS** – both male and female sexes eligible for recruitment<br>• **Interpretation** – effects of CCR5 antagonists for stroke recovery in female animals represents a critical knowledge gap |
| Animals with comorbidities | No | No | • **Preclinical** – 0/5 studies included animals with common stroke comorbidities (e.g., diabetes, hypertension, etc.)<br>• **CAMAROS** – participants with stroke comorbidities eligible for recruitment<br>• **Interpretation** – effects of CCR5 antagonists for stroke recovery in animals with common stroke comorbidities represents a critical knowledge gap |
| Evidence from two or more laboratories | Yes | No | • **Preclinical** – 5/5 studies demonstrated stroke-related benefits; however, no study had their results replicated by independent laboratories<br>• **CAMAROS** – trial protocol will be replicated by multiple study sites<br>• **Interpretation** – future preclinical experiments could consider assessing effects using a multilaboratory approach, or independently replicate the effects of existing preclinical studies |
| Gyrencephalic species | No | No | • **Preclinical** – 5/5 studies conducted in lissencephalic (rodent) species.<br>• **CAMAROS** – human participants<br>• **Interpretation** – effects of CCR5 antagonists for stroke recovery in gyrencephalic species represents a critical knowledge gap |
| Tests during the awake phase of animals | No | No | • **Preclinical** – 0/5 studies reported assessing outcomes at night (i.e., awake phase of rodents)<br>• **CAMAROS** – follow-up occurs during daytime hours (i.e., awake phase of humans)<br>• **Interpretation** – future preclinical studies should conduct behavioural testing during the awake phase to better align with human testing conditions |

for the effects of CCR5 antagonists in clinically relevant sexes, ages, species, and disease comorbidities, without sufficient dose–response information to inform trials (*Appendix 2—table 1*). Overall, our assessments highlight a variety of knowledge gaps, potential confounding factors, and areas of misalignment between the preclinical evidence and clinical trial parameters that could be improved with further preclinical experimentation.

## Discussion

The overall body of preclinical evidence for CCR5 antagonists in stroke demonstrates potential acute cerebroprotection with corresponding impairment reduction, as well as improved functional recovery in the subacute/early chronic phase. Our systematic review also highlights evidence gaps that could impact successful clinical translation of CCR5 antagonists. Our analysis of 10 independent experiments, identified that acute administration of CCR5 antagonists within the first 24-hr post-stroke was associated with a marked reduction in infarct volume. This cerebroprotective reduction of infarct volume did not significantly vary based on treatment dose or any other experimental characteristics (*Takami et al., 2002*; *Chen et al., 2022*; *Joy et al., 2019*). Overall, the majority of behavioural effects appeared to be in a positive direction, but the low number of included studies precluded meta-analysis of these results. Indeed, no individual behavioural experiment included more than 10 CCR5-treated animals, and given the wide range of dosages, timings, routes, stroke models, and rodent strains involved, the certainty of these findings is limited and should be interpreted cautiously. Pooling

**Table 4.** Alignment of included preclinical studies with additional STAIR/SRRR items and CAMAROS trial parameters.

| Recommendation | Preclinical evidence | CAMAROS alignment | Notes |
|---|---|---|---|
| Testing in both permanent and transient occlusion models | Yes | Yes | • **Preclinical** – 2/5 studies permanent occlusion, 2/5 transient occlusion, 1/5 haemorrhagic<br>• **CAMAROS** – participants with ischaemic anterior circulation stroke, with or without reperfusion treatments eligible for enrollment |
| Monitoring of treatment effects on physiological parameters, including temperature, both during and after ischaemia | No | NA | • **Preclinical** – 1/5 studies (*Takami et al., 2002*; TAK-779) monitored and provided data on temperature for an extended period post-stroke<br>• **CAMAROS** – participants will not be administered maraviroc in a cerebroprotective context |
| Testing interaction with thrombolytics and other medications commonly administered in acute stroke | No | No | • **Preclinical** – 0/5 studies included assessed drug interactions<br>• **CAMAROS** – participants may or may not be exposed to thrombolytics or other concomitant medications that could interact with recovery-inducing effects of maraviroc |
| Animal model should produce infarction similar in relative size and location to that observed in humans | Yes | Yes | • **Preclinical** – 1/5 studies (*Joy et al., 2019*; maraviroc) produced infarcts in motor regions of control animals that were less than 25% of hemispheric volume<br>• **CAMAROS** – participants with ischaemic anterior circulation stroke with hemiparesis requiring rehabilitation eligible for enrollment |
| For studies claiming recovery effects, analysis of infarct volume should be performed to show equivalency of injury in the treated and stroke control groups | No | NA | • **Preclinical** – 0/1 experiments in late subacute/early chronic post-stroke phase (*Joy et al., 2019*) demonstrated that each treatment group had equivalent baseline stroke volumes. This could potentially confound observed behavioural effects |
| For studies claiming recovery effects, tissue repair/neuroplasticity processes should be quantified and directly related to behavioural recovery | Yes | NA | • **Preclinical** – 1/5 studies (*Joy et al., 2019*) demonstrated maraviroc association with dendritic spine preservation, axonal projections to contralateral cortex, reduced inflammatory response, and upregulation of CREB/DLK signalling<br>• **CAMAROS** – outcomes to assess mechanism of recovery are not included |
| For studies claiming recovery effects, initial impairment with spontaneous recovery that plateaus significantly below pre-stroke performance | No | NA | • **Preclinical** – 1/5 studies assessed long-term outcomes (*Joy et al., 2019*). Large variability in initial levels of impairment across experiments. Control animals demonstrate limited spontaneous recovery or atypical worsening of performance across time in multiple experiments.<br>• **CAMAROS** – placebo participants will receive 8-week exercise program making assessment of spontaneous recovery difficult |
| Behavioural effects should be assessed across a battery of domains, including both skilled and spontaneous upper limb and hindlimb use | No | No | • **Preclinical** – 2/5 studies included battery of behavioural tasks, but no study assessed effects on upper extremity skilled reaching/grasping or alteration of movement kinematics<br>• **CAMAROS** – two primary outcomes: (1) upper extremity assessment using Fugl-Meyer; (2) change in 10-meter walk test score. Preclinical outcomes have limited relevance to Fugl-Meyer |
| Mechanism of action should be assessed through gain of function/loss of function studies and directly associated to behavioural effects | Yes | NA | • **Preclinical** – 1/5 studies *Joy et al., 2019* demonstrated that knockdown of DLK signalling using small hairpin DLK is able to block beneficial behavioural effects of CCR5 knockdown, providing a possible mechanism for stroke recovery-inducing effec<br>• **CAMAROS** – outcomes to assess mechanism of recovery are not included |
| Testing interactions of treatment with clinically inspired best practice, such as training or enrichment | No | No | • **Preclinical** – 0/5 studies included behavioural therapy/rehabilitative interventions<br>• **CAMAROS** – all participants (including placebo) will participate in an 8-week exercise program |

data across heterogenous experimental designs, animal/stroke models, and treatment parameters, as we have done with the infarct volume analysis in the present study, can introduce variability that increases the risk of overestimating or underestimating the true effect of the intervention (*Higgins et al., 2023b*). Treatment effects observed across model systems and therapeutic compounds may represent different biological mechanisms. Despite this potential limitation, meta-analysis can provide valuable insights, especially in preclinical settings where the sample sizes of individual studies may be too small to detect significant effects on their own. In these cases, pooling data across studies can help identify overarching estimates of benefits and harm, highlight subgroups of interest, and help guide areas of future research. As described in the results above, we attempted to mitigate the risks of inappropriate data pooling through careful investigation of heterogeneity, subgroup analyses, and differentiation between outcomes where we felt that meta-analytic pooling was (infarct volume) and was not (behavioural outcomes) appropriate. Overall, we believe that our results indicate that further investigation is warranted to determine the optimal timing of administration and behavioural domains under which CCR5 antagonists exhibit the strongest post-stroke cerebroprotective and recovery-inducing effects.

Despite the positive direction of treatment effects across all studies of CCR5 antagonists, we found a substantial risk of bias in the underlying studies (*Hooijmans et al., 2014*). As is commonly observed in the preclinical literature, all studies either did not adequately report their randomization/blinding methods and exhibited evidence of selective/incomplete reporting (*Avey et al., 2016*; *Fergusson et al., 2019b*; *Fergusson et al., 2019a*). Such features are associated with biased overestimations of preclinical treatment efficacy, which raises further concerns about the reliability and validity of the present findings (*Holman et al., 2016*; *Sena et al., 2010*; *Macleod et al., 2008*). Future studies should carefully incorporate all elements of the ARRIVE 2.0 guidelines to help ensure that results are transparently reported and improve confidence in the findings (*Percie du Sert et al., 2020*).

Comprehensiveness of the preclinical evidence for CCR5 antagonists was assessed in relation to STAIR and SRRR consensus recommendations (*Corbett et al., 2017*; *Lyden et al., 2021*; *Finklestein et al., 1999*; *Fisher et al., 2009*). These recommendations aim to provide investigators and regulators with 'assurance that the candidate treatment shows signals of efficacy and safety, before embarking on an expensive clinical development program' (*Lyden et al., 2021*). The included studies provide good initial evidence for acute cerebroprotection, as well as mechanistic and behavioural evidence for enhanced recovery in the late subacute/early chronic post-stroke phase. However, demonstration of efficacy under a wider range of conditions, such as in aged animals, females, animals with stroke-related comorbidities, more clinically relevant timing of dose administrations, or in conjunction with rehabilitative therapies are necessary to provide further confidence in these findings. In addition, all studies used unique doses, timings, and outcomes, so independent replication of the most promising study parameters would further increase certainty in the evidence (*Corbett et al., 2017*; *Lyden et al., 2021*). Future preclinical research should aim to address these evidence gaps to further increase the clinical relevance and comprehensiveness of evidence for CCR5 antagonists in stroke.

In relation to the ongoing CAMAROS trial assessing maraviroc in the subacute post-stroke phase (*Dukelow, 2023*), the most relevant preclinical evidence comes from one experiment within the Joy et al. study (*Joy et al., 2019*) where maraviroc was initially administered at 3–4 weeks post-stroke and continued daily for 11 weeks. This experiment demonstrated that administration of maraviroc in the late subacute/early chronic post-stroke phase improved functional recovery on the grid walk, but not cylinder, task. These experimental conditions could potentially align with the putative therapeutic window and outcomes of interest being assessed in CAMAROS (e.g., 10-min walk test co-primary outcome) (*Dukelow, 2023*). However, caution is warranted as this pivotal supporting preclinical evidence is based on a low sample size ($n = 9$). Moreover, potential differences in dosing, severity of infarct, concomitant rehabilitative therapy, and other factors discussed above could influence the degree to which these results successfully translate to the clinical environment. Finally, clinically relevant secondary outcomes in the CAMAROS trial, such as cognitive and emotional domains as measured by the Montreal Cognitive Assessment (MoCA) and Stroke Aphasia Depression Questionnaire (SADQ) were not modelled in the preclinical literature. Although one study included a cognitive outcome, the other treatment parameters of this study were not aligned to the CAMAROS trial (*Yan et al., 2021*). Future preclinical studies should assess a more diverse and comprehensive battery of clinically relevant behavioural tasks, which could be based on the range of outcomes employed in the

CAMAROS trial, or those found in the SRRR recommendations (*Corbett et al., 2017*). Nevertheless, the Joy et al. study provides a plausible biological mechanism and 'proof of concept' for how CCR5 antagonism might enhance neuroplasticity that improves functional recovery after stroke.

Our present synthesis of the preclinical evidence for CCR5 antagonists used novel approaches to increase its utility for assessing certainty of the findings and identification of knowledge gaps. First, we engaged patients with lived experiences of stroke throughout the review process to ensure that our research questions, outcomes, and interpretations aligned with the priorities of the ultimate end-user of stroke research. Second, we incorporated consensus recommendations for both preclinical cerebroprotection and recovery research as an evidence evaluation tool, which we found often aligned with the priorities of our patient panel (*Corbett et al., 2017*; *Lyden et al., 2021*; *Finklestein et al., 1999*; *Fisher et al., 2009*). This guided assessment of the alignment of preclinical evidence with parameters of ongoing clinical trials, as well as appraisal of comprehensiveness of the preclinical evidence with a focus on translational validity (*Drude et al., 2021*) rather than only internal validity and risk of bias (*Hooijmans et al., 2014*). We also used this method to provide concrete avenues for future preclinical studies to close knowledge gaps and improve certainty in the effects of CCR5 antagonists under clinically relevant experimental conditions. Similar approaches should be considered by future preclinical systematic reviews to improve interpretation of the preclinical evidence from a translational perspective.

In conclusion, CCR5 antagonists show promise in preclinical studies for stroke cerebroprotection, corresponding reduction in impairment, as well as improved functional recovery related to neural repair in the late subacute/early chronic phase. However, high risk of bias and the limited (or no) evidence in clinically relevant domains underscore the need for more rigorous and transparent preclinical research to further strengthen the current preliminary evidence available in the literature. Addressing these concerns will not only enhance the reliability of preclinical evidence but also better inform the design and execution of clinical trials of CCR5 antagonists, such as the ongoing CAMAROS trial (*Dukelow, 2023*). The integration of expert recommendations, such as STAIR and SRRR, should guide future preclinical investigations and synthesis of the body of preclinical evidence in stroke recovery research (*Corbett et al., 2017*; *Lyden et al., 2021*). Our present approach serves as a template by which the preclinical evidence supporting translation to clinical trials can be weighed when justifying early clinical trials of novel interventions for stroke cerebroprotection and recovery.

## Materials and methods

We registered the review protocol on the International Prospective Register of Systematic Reviews (CRD42023393438) (*Lalu et al., 2023*). The findings are reported in accordance with the Preferred Reporting Items for Systematic Reviews and Meta-Analyses (see attached Reporting Standards Document) (*Page et al., 2021*) and Guidance for Reporting the Involvement of Patients and the Public (*Appendix 3—table 1*; *Staniszewska et al., 2017*).

### Engagement with patient panel – individuals with lived experience of stroke

A panel of eight patients and caregivers with lived experience of a stroke informed project development (e.g., research question development, review protocols, search strategy development) and were actively involved in the research conduct (screening, data extraction, analysis, interpretation). Monthly meetings occurred with the patients and caregivers to provide educational sessions of background knowledge of preclinical stroke, systematic review conduct, and discuss research findings as the review progressed. We co-developed a terms of reference document a priori to document details of the engagement (i.e., roles, responsibilities, expectations, project goals, etc.). The patient partners co-developed the research question to address patient interests, including chronic stroke recovery (i.e., the panel was particularly interested in evaluating the effects of extended drug administration), consideration of physical therapy in tandem with drug administration, inclusion of stroke-relevant comorbidities, spasticity, and motor and cognitive outcomes. Co-authorship and financial compensation were agreed upon with the patients and caregivers and offered as a method of acknowledgement according to the Canadian Institutes of Health Research (CIHR) Strategies for Patient Oriented Research (SPOR) Evidence Alliance Patient Partner Appreciation Policy.

## Eligibility criteria

- *Animals*: Any preclinical in vivo animal models of adult stroke were included. Human, invertebrate, in vitro, ex vivo, and neonatal animal studies were excluded.
- *Model*: Focal ischaemic or intracerebral haemorrhagic stroke models were included, while animal models of global ischaemia were excluded.
- *Intervention*: Studies administering a CCR5 antagonist drug (e.g., maraviroc, D-Ala-Peptide T-Amide [DAPTA], Takeda 779 [TAK-779]). Study arms in which CCR5 was genetically manipulated (e.g., CCR5 knockout strain) were excluded.
- *Comparator:* Vehicle-treated control groups where stroke was induced. CCR5 antagonist control groups without stroke were excluded.
- *Outcome*: Studies reporting at least one of the following: infarct size, behavioural tests, mortality, adverse events, and spasticity were included.
- *Study design and publication characteristics*: Controlled interventional studies (randomized, pseudo-randomized, or non-randomized) published as full journal articles in any language or year were included. Abstracts, review articles, opinion-based letters/editorials, and unpublished grey literature were excluded.

## Information sources and search strategy

An information specialist with experience in systematic searches of the preclinical literature developed a comprehensive search strategy based on a previously published strategy for identifying animal experimentation studies (*Supplementary file 1*; *Hooijmans et al., 2010*). The search strategy underwent peer-review using the Peer Review of Electronic Search Strategies (PRESS) checklist (*McGowan et al., 2016*). We searched MEDLINE (OVID interface, including In-Process and Epub Ahead of Print), Web of Science, and Embase (OVID interface). The search was conducted 25 October 2022, to align with the listed launch date of the CAMAROS trial (15 September 2022). Our intention in doing so was to collate and assess all preclinical evidence that could have feasibly informed the clinical trial. We sought to assess the comprehensiveness of evidence and readiness for translation of CCR5 antagonist drugs at the time of their actual translation into human clinical trials, as well as the alignment of the CAMAROS trial design to the existing preclinical evidence base.

## Selection process and data collection

We deduplicated citations and uploaded them into DistillerSR (Evidence Partners, Ottawa, Canada). Two reviewers independently screened citations by title and abstract using an accelerated method (one reviewer required to include, two reviewers required to exclude). We then screened and extracted data from full-text articles in duplicate. Graphical data was extracted using Engauge Digitizer (*Mitchell et al., 2023*). A third reviewer with content expertise in preclinical stroke studies audited all data extraction. Conflicts between reviewers were resolved by consensus discussion. See *Supplementary file 2* for the complete list of data extraction elements.

## Effect measures and data synthesis

We performed quantitative analyses using the R (version 4.1.2) 'metafor' package (version 4.0.0) (*Viechtbauer, 2010*) with inverse variance random effects modelling. We expressed continuous outcome measures as SMDs with 95% CIs and assessed statistical heterogeneity of effect sizes using the Cochrane $I^2$ statistic (*Higgins et al., 2023a*). This was necessary due to the variety of outcome measures and measurement scales used across studies, which is a common feature of preclinical systematic reviews. Sensitivity analyses were performed using original measurement scales where possible. From patient partner input, subgroups were analysed based on timing/dose/route of intervention, stroke model, stroke type, species, type of behavioural outcome (i.e., motor, cognitive), and comorbidities (*Bernhardt et al., 2017*). Our planned subgroup analyses on study quality, specific regions/areas of the brain, and post-stroke rehabilitation paradigms were not performed due to insufficient number of studies. We did not assess publication bias using Egger plots due to fewer than 10 studies being included in the analysis, as per Cochrane recommendations (*Page et al., 2023*).

## Risk of bias assessment

Two independent reviewers used the SYRCLE risk of bias tool to assess each study as having a 'Low Risk', 'Unclear Risk', 'Some Concerns of Risk', or 'High Risk' across domains such as randomization, blinding, and outcome reporting (*Hooijmans et al., 2014*) 'Some Concerns of Risk' indicated reporting of a domain (i.e., randomization), but lacking methodological details. This differed from 'Unclear Risk' where there was no mention of the domain in the study.

## Comprehensiveness of preclinical evidence and alignment with clinical trials

We assessed comprehensiveness of the *overall body* of preclinical evidence for CCR5 antagonists as a cerebroprotective treatment using The Stroke Treatment Academic Industry Roundtable (STAIR) I, VI, and XI Consolidated Recommendations (*Lyden et al., 2021*; *Finklestein et al., 1999*; *Fisher et al., 2009*). These recommendations encompass 'candidate treatment qualification' (e.g., dose, timing of dose, outcomes) and 'preclinical assessment and validation' (e.g., age, sex, sample size, animal type) (*Lyden et al., 2021*). We excluded domains that were redundant with risk of bias (e.g., randomization, blinding, etc.) and included additional items relevant to stroke recovery studies from the Stroke Recovery and Rehabilitation Roundtable (SRRR) Translational Working Group consensus recommendations (*Corbett et al., 2017*). Two reviewers extracted data to determine if the overall evidence across studies satisfied each of the STAIR and SRRR recommendations. A third reviewer audited this analysis. We then assessed alignment of existing preclinical evidence with an ongoing clinical trial of CCR5 antagonists for stroke (CAMAROS, NCT04789616) (*Dukelow, 2023*).

## Protocol deviations

A list of outcomes used to determine CCR5's potential mechanisms of action were extracted (*Appendix 1—figure 1*, *Appendix 1—table 1*) based on feedback from individuals that reviewed the initial manuscript drafts. We also incorporated an additional assessment using the PRIMED (*Li et al., 2016*) tool for assessing the readiness of stroke cerebroprotection therapies to be translated to clinical trials based on the body of preclinical evidence (*Appendix 2—table 1*; *Bahr-Hosseini et al., 2022*).

## Acknowledgements

The authors thank Hannah Laquerre for her assistance with figure generation. This study was funded by a Social Accountability Grant from the University of Ottawa's Faculty of Medicine. MSJ was supported by the Canadian Institutes of Health Research (CIHR) and Vanier Canada Graduate Scholarship (CGV-186957). The funders were not involved in the study design, collection, analysis, and interpretation of data, writing the manuscript, or in the decision to submit the article for publication. MML is supported by The Ottawa Hospital Anesthesia Alternate Funds Association, a University of Ottawa Junior Research Chair, and the Canadian Anesthesiologists' Society Career Scientist Award.

## Additional information

### Funding

| Funder | Grant reference number | Author |
| --- | --- | --- |
| University of Ottawa | Social Accountability Grant | Ayni Sharif<br>Dean A Fergusson<br>Manoj M Lalu |
| Canadian Institutes of Health Research - Vanier Canada Graduate Scholarship | CGV-186957 | Matthew S Jeffers |
| Ottawa Hospital Anesthesia Alternate Funds Association | | Manoj M Lalu |

| Funder | Grant reference number | Author |
|---|---|---|
| Canadian Anesthesiologists' Society | | Manoj M Lalu |

The funders had no role in study design, data collection and interpretation, or the decision to submit the work for publication.

**Author contributions**

Ayni Sharif, Conceptualization, Data curation, Software, Formal analysis, Funding acquisition, Validation, Investigation, Visualization, Methodology, Writing – original draft, Project administration, Writing – review and editing; Matthew S Jeffers, Conceptualization, Resources, Data curation, Software, Formal analysis, Supervision, Validation, Investigation, Visualization, Methodology, Writing – original draft, Project administration, Writing – review and editing; Dean A Fergusson, Manoj M Lalu, Conceptualization, Resources, Supervision, Funding acquisition, Methodology, Writing – original draft, Writing – review and editing; Raj Bapuji, Investigation, Writing – review and editing; Stuart G Nicholls, Conceptualization, Resources, Methodology, Writing – review and editing; John Humphrey, Warren Johnston, Ed Mitchell, Mary-Ann Speirs, Laura Stronghill, Michele Vuckovic, Susan Wulf, Conceptualization, Investigation, Methodology, Writing – review and editing; Risa Shorr, Resources, Software, Investigation, Methodology, Writing – review and editing; Dar Dowlatshahi, Conceptualization, Writing – review and editing; Dale Corbett, Conceptualization, Resources, Investigation, Methodology, Writing – original draft, Writing – review and editing

**Author ORCIDs**
Matthew S Jeffers ⓘD https://orcid.org/0000-0002-4148-2638
Manoj M Lalu ⓘD https://orcid.org/0000-0002-0322-382X

Joint Public Review: https://doi.org/10.7554/eLife.103245.3.sa1
Author response https://doi.org/10.7554/eLife.103245.3.sa2

# Additional files

**Supplementary files**
Supplementary file 1. Search strategy.
Supplementary file 2. Data extraction forms.
MDAR checklist

**Data availability**

All data generated or analyzed during this study are included in the manuscript and supporting files. Source data and code to generate Figures 2 and 3 have been provided at the following DOI: https://doi.org/10.17605/OSF.IO/5KGT6.

The following dataset was generated:

| Author(s) | Year | Dataset title | Dataset URL | Database and Identifier |
|---|---|---|---|---|
| Sharif A, Jeffers MS, Fergusson DA, Bapuji R, Nicholls SG, Humphrey J, Johnston W, Mitchell E, Speirs MA, Stronghill L, Vuckovic M, Wulf S, Shorr R, Dowlatshahi D, Corbett D, Lalu MM | 2025 | CCR5 antagonists as cerebroprotective and stroke recovery enhancing agents: a preclinical systematic review and meta-analysis - eLife datasets | https://doi.org/10.17605/OSF.IO/5KGT6 | Open Science Framework, 10.17605/OSF.IO/5KGT6 |

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

## Appendix 1

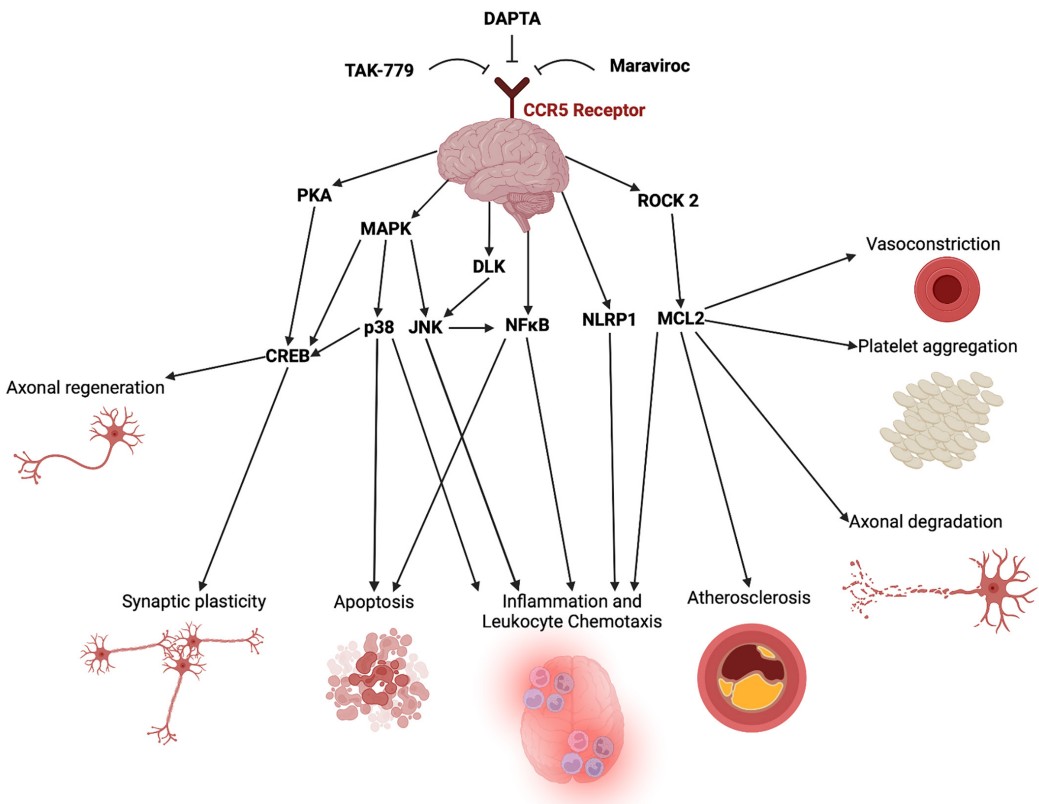

**Appendix 1—figure 1.** Potential mechanistic pathways and proposed domains of biological activity underlying CCR5 antagonist's cerebroprotective and neural repair effects post-stroke described in the included studies. A list of mechanistic evidence supporting these pathways was extracted from included studies in **Appendix 1—table 1**. This figure was created using BioRender.com.

**Appendix 1—table 1.** List of outcomes used to determine potential mechanisms of action, by study.

| Paper | Mechanistic evidence presented |
|---|---|
| *Takami et al., 2002* – TAK-779 | • Infarct volume reduction (reduced by TAK-779)<br>• MPO + Mac-1-alpha antibody used to stain infiltrating neutrophils/macrophages/microglia (Mac-1-alpha reduced by TAK-779/MPO not decreased) – neutrophils the same, macrophages/microglia reduced |
| *Li et al., 2016* – DAPTA | • Cell death reduction (TUNEL/H&E)<br>• Western blot – ROCK2/P-MLC2 reduced (by DAPTA) |
| *Joy et al., 2019* – maraviroc | • CCR5 knockdown FACS isolated neuronal assay -> CREB/pCREB/DLK significantly increased. Erk(p44), GAP43, SAP/JNK1+2 increased, but not significantly. P38 decreased, but not significantly<br>• Two-photon imaging – CCR5 knockdown + maraviroc -> increased dendritic spine count, fewer lost<br>• BDA axonal quantification – CCR5 knockdown + maraviroc -> increased bihemispheric sprouting<br>• GFAP/IBA-1 immunostaining – CCR5 knockdown + maraviroc -> reduced immunoreactivity. Decreased Ly6C (neutrophil) and Ly6Clow (macrophage) recruitment |

*Appendix 1—table 1 Continued*

| Paper | Mechanistic evidence presented |
|---|---|
| ***Yan et al., 2021*** *– maraviroc* | • Cell death reduction (TUNEL/FJC/cleaved caspase 1/Nissl)<br>• Western blot – maraviroc -> increased PK A-Calpha/pCreb -> decreased NLRp1/cleaved caspase 1/IL-1B/apoptosis-associated speck protein/N-gasdermin D/IL-18<br>• Western blot -> effects reversed by 666-15<br>• Western blot -> rCCL5 shows reversed effects from maraviroc -> rCCL5 + cAMP reverses those effects |
| ***Chen et al., 2022*** *– maraviroc* | • Infarct volume/cell death reduction (TTC/TUNEL)<br>• Western blot – maraviroc -> increased Bcl2:BAX ratio -> increased IκBα, ->decreased P-IκBα/P-P65/P-P38/P-JNK<br>• Western blot – P-JNK decreased, Anisomycin reverses the effect on P-JNK<br>qRT-PCR, ELISA -> reduced IL-1B/IL-6/TNF-alpha |

# Appendix 2.

**Appendix 2—table 1.** Readiness of CCR5 antagonists for translation based on the PRIMED2 tool.

| PRIMED2 domain | TAK-779 | DAPTA | Maraviroc | CCR5 antagonism (all agents) |
|---|---|---|---|---|
| Sex of animals<br>(0 – male OR female; 2 – both sexes) | 0 | 0 | 0 | 0 |
| Age of animals<br>(0 – young only; 2 – older adults) | 0 | 0 | 0 | 0 |
| Species and strains<br>(0 – one rodent species/strain;<br>1 – ≥2 rodent species/strains;<br>2 – rodents AND primates) | 0 | 0 | 1 | 1 |
| Reproducibility<br>(0 – one species, one laboratory;<br>1 – ≥2 species, one lab OR ≥2 labs, one species;<br>2 – ≥2 species AND ≥2 labs) | 0 | 0 | 2 | 2 |
| Treatment time epoch<br>(0 – no significant benefit;<br>1 – benefit in one epoch;<br>2 – benefit in ≥2 epochs) | 1 | 1 | 2 | 2 |
| Baseline comorbidities<br>(0 – healthy animals only;<br>1 – one comorbid condition;<br>2 – ≥2 comorbid conditions) | 0 | 0 | 0 | 0 |
| Feasible time window<br>(0 – treatment <45 min after ischemia onset;<br>2 – ≥45 min after ischemia onset) | 0 | 0 | 2 | 2 |
| Dose–response<br>(0 – one dose;<br>2 – multiple doses with dose–response relationship) | 0 | 0 | 0 | 0 |
| Feasible route of delivery<br>(0 – infeasible route [e.g., intraventricular injection];<br>1 – semi-infeasible route [e.g., intraarterial injection];<br>2 – feasible route [e.g., intravenous injection]) | 2 | 1 | 1 | 2 |
| Behavioural and/or long-term outcome assessment<br>(0 – no benefit;<br>1 – behavioural OR other outcome ≥30 days post-stroke;<br>2 – behavioural AND outcomes ≥30 days post-stroke) | 0 | 0 | 2 | 2 |
| Typical infarct volume reduction magnitude<br>0 – small effect (Cohen's d 0.2–0.39);<br>1 – medium effect (Cohen's d 0.4–0.69);<br>2 – large effect (Cohen's d≥0.7) | 2 | 2 | 2 | 2 |
| Readiness for translation score<br>(0–7 – low; 8–15 – medium; 16–22 – high) | Low (5) | Low (4) | Medium (12) | Medium (13) |

# Appendix 3.

**Appendix 3—table 1.** GRIPP2 short form checklist.

| Section and topic | Item |
|---|---|
| 1: Aim | To conduct a preclinical systematic review assessing the effects of C-C chemokine receptor type 5 (CCR5) antagonists on motor and cognitive impairment following stroke. To collaborate with a panel of patients and caregivers with lived experience of stroke throughout the development and conduct of the preclinical systematic review. |
| 2: Methods | A panel of eight patients and caregivers with lived experience of stroke was recruited to join the research team through the Heart & Stroke Foundation and the Patient and Family Advocacy Program at The Ottawa Hospital. Recruitment advertisements were distributed to both organizations. The patients and caregivers were involved in developing the research question and the protocol (i.e., elements of the AMICO question presented in the Methods section); identifying data items for extraction; conduct of the review including screening and extraction; interpreting the review results; and contributed to edits to the final manuscript. Patients and caregivers attended monthly virtual meetings. These meetings provided background knowledge of preclinical stroke, systematic review conduct, and discussed research findings as the review progressed. Patients provided insights into their lived experiences with stroke and helped identify priority areas that they were interested in. Patients and caregivers were offered financial compensation and co-authorship in recognition of their contributions to the research project. |
| 3: Results | Patient engagement contributed to the study in several ways, including:<ul><li>Informing the research question with the patient partner experience: patient partners have lived experience of stroke and provided patient priorities to analyze in our review</li><li>Refining our primary and secondary outcomes to include patient priorities</li><li>Developing of the protocol to include non-technical language</li><li>Editing screening and extraction forms</li><li>Participating as reviewers in abstract and full-text level screening</li><li>Participating as reviewers in extraction</li><li>Attending an international stroke conference and participating as co-authors of an abstract presented there</li><li>Interpretating analysed data, including identifying patient important barriers and limitations to incorporate in the discussion</li></ul> |
| 4: Discussion | Overall, patient engagement was successful in informing review development and conduct. Additionally, the researchers on the team learned from the patient panel. The patient panel brought a unique perspective to the planning and conduct of a preclinical systematic review. Additionally, members of the panel stated that the experience was useful for them as they gained new insights into preclinical research. |

*Appendix 3—table 1 Continued on next page*

*Appendix 3—table 1 Continued*

| Section and topic | Item |
| --- | --- |
| 5: Reflections | Engagement was embedded within the research project from the inception to dissemination, where patient partners were members of the larger team. Patient partners directed us to outcomes for the systematic review that critical clinically to patients in the chronic phase of stroke recovery. Their participation in the conduct of the review facilitated meaningful collaborations and discussion. |

