## [Editor Report · eLife Assessment]

This study is **important**, and the findings add substantially to the evidence base regarding CCR5 antagonist drugs for neuroprotection and stroke management. The authors adhered to the expected systematic review and meta-analysis standards, and the presented evidence is **convincing**.

---

## [Referee Report · Joint Public Review]

This is an interesting, timely, and high-quality study on the potential neuroprotective capabilities of C-C chemokine receptor type 5 (CCR5) antagonists in ischemic stroke. The focus is on preclinical investigations.

An outstanding feature is that stroke patient representatives have directly participated in the work. Although this is often called for, it is hardly realized in research practice, so the work goes beyond established standards.

The included studies were assessed regarding the therapeutic impact and their adherence to current quality assurance guidelines such as STAIR and SRRR, another important feature of this work. While overall results were promising, there were some shortcomings regarding guideline adherence.

The paper is very well written and concise yet provides much highly useful information. It also has very good illustrations, and extremely detailed and transparent supplements.

[Editors' note: The authors have responded appropriately to the comments shared by the reviewers. The authors have provided a good academic justification for not needing to update the literature search, as one of the reviewers had suggested.]

---

## [Author Response]

The following is the authors’ response to the original reviews.

**Public Reviews:**

**Reviewer #1 (Public review):**
Summary:The paper is well-organized, with clearly defined sections. The systematic review methodology is thorough, with clear eligibility criteria, search strategy, and data collection methods. The risk of bias assessment is also detailed and useful for evaluating the strength of evidence. The involvement of a patient panel is noticeable and positive, ensuring the research addresses real-world concerns and aligning scientific inquiry with patient perspectives. The statistical approach used for analyzing seems appropriate.The authors are encouraged to take into account the following points:As the authors have acknowledged, there is a high risk of bias across all included studies, particularly in randomization, selective outcome reporting, and incomplete data, which could be highlighted more explicitly in the paper's discussion section, particularly the potential implications for the generalizability of the results. The authors can also suggest mitigation strategies for future studies (e.g., better randomization, blinding, reporting standards, etc.).

We agree that it is important to highlight mitigation strategies that will allow preclinical researchers to more transparently report future studies. We have directed readers to ensure reporting in alignment with the ARRIVE 2.0 guidelines for further details on reporting of preclinical studies, as follows in paragraph two of the Discussion, “Future studies should carefully incorporate all elements of the ARRIVE 2.0 guidelines to help ensure that all results are transparently reported and improve confidence in the findings.(41)”

None of the studies include female animals, and the use of young adult animals (instead of aged models) limits the applicability of the findings to the human stroke population, where stroke incidence is higher in older adults and perhaps the gender issue must be included to reflect the translational aspects. The authors can add to the paper's discussion section that perhaps future preclinical studies should include both sexes and aged animals to align better with the clinical population and improve the translation of findings. Another point is the comorbidity. Comorbidities such as diabetes and hypertension are prevalent in stroke patients. How can these be considered in preclinical designs? The authors should emphasize the importance of future research incorporating such comorbid models to enhance clinical relevance. None of the studies had independent replication of their findings, which is a key limitation, especially for a field with high translational expectations. This should be highlighted as a critical next step for validating the efficacy of CCR5 antagonists.

We agree that these are important evidence gaps to address. Although we highlighted these gaps in paragraph 3 of the Discussion, we have now added a more explicit call to action for researchers to address these gaps at the end of the relevant paragraph as follows, “Future preclinical research should aim to address these evidence gaps to further increase the clinical relevance and comprehensiveness of evidence for CCR5 antagonists in stroke.”

The studies accessed limited cognitive outcomes (only one reported a cognitive outcome). Given the importance of cognitive recovery post-stroke, this is a gap to highlight in the discussion. Future studies should include more diverse and comprehensive behavioral assessments, including cognitive and emotional domains, to fully evaluate the therapeutic potential.

We have expanded on this important point in paragraph four of the Discussion, which explores the alignment of the preclinical literature to the CAMAROS trial, as follows, “Finally, clinically relevant secondary outcomes in the CAMAROS trial, such as cognitive and emotional domains as measured by the Montreal Cognitive Assessment (MoCA) and Stroke Aphasia Depression Questionnaire (SADQ) were not modelled in the preclinical literature. Although one study included a cognitive outcome, the other treatment parameters of this study were not aligned to the CAMAROS trial. Future preclinical studies should assess a more diverse and comprehensive battery of clinically relevant behavioural tasks, which could be based on the range of outcomes employed in the CAMAROS trial, or those found in the SRRR recommendations.(9)”

This addition highlights the lack of supporting preclinical evidence for cognitive recovery post-stroke. We also offer recommendations on discrete ways to address this gap in future preclinical studies by taking inspiration from the outcomes used in CAMAROS as well as the SRRR guidelines used throughout our assessment of the CCR5 literature.

The timing of CCR5 administration across studies varies widely (from pre-stroke to several days post-stroke) complicating the interpretation and comparison of results. The authors are encouraged to add that future preclinical studies could focus on narrowing the therapeutic window to more clinically relevant time points.

We agree with the review and feel that this recommendation is currently captured in paragraph three of our Discussion - “However, demonstration of efficacy under a wider range of conditions, such as in aged animals, females, animals with stroke-related comorbidities, more clinically relevant timing of dose administrations, or in conjunction with rehabilitative therapies are necessary to provide further confidence in these findings.” As mentioned above, we added a new sentence to the end of this paragraph to make it more explicit that these are gaps that should be addressed by future preclinical research. “Future preclinical research should aim to address these evidence gaps to further increase the clinical relevance and comprehensiveness of evidence for CCR5 antagonists in stroke.” We also added the word “clinically” to the original sentence mentioned above to more explicitly align with the reviewer’s recommendation.

The paper identifies some alignment with clinical trials, but there are several gaps, too, particularly in the types of behavioral tests used in preclinical studies versus those in clinical trials. If this systematic review and meta-analysis aim to formulate a set of recommendations for future studies, it is important that the authors also propose specific preclinical behavioral tasks that could better align with clinical measures used in trials, like functional assessments related to human stroke outcomes.

As mentioned above, we added a sentence to Discussion paragraph four, the comparison to the CAMAROS trial, that provides recommendations as to the behavioural tasks that would be useful to employ in future studies. Namely, “Future preclinical studies should assess a more diverse and comprehensive battery of clinically relevant behavioural tasks, which could be modelled after the range of outcomes employed in the CAMAROS trial, or those found in the SRRR recommendations.(9)” The SRRR recommendations that we reference here provide discrete consensus recommendations for interested readers on behavioural task selection, as well as priority rankings based on rodent species, to better align with clinical measures used in trials.

The discussion needs some revisions. It could benefit from an expanded explanation of CCR5's mechanistic role in neuroplasticity and stroke recovery. For instance, linking CCR5 antagonism more closely with molecular pathways related to synaptic repair and remyelination would enhance the quality of the discussion and understanding of the drugs' potential.

We have provided a synthesis of CCR5’s proposed mechanistic roles in the Supplementary Materials, Figure S1 (for a summary pathway diagram), and Table S3 (for a list of potential mechanistic pathways and supporting evidence presented in each paper). Given our focus on study quality and alignment with translational recommendations, we felt that it was more appropriate to not focus on mechanistic elements in the Discussion. Indeed, the appraisal of the quality of support for each potential mechanism was beyond the scope of our present analysis.

While the tool is used to assess the risk of bias, it might be helpful to integrate a broader framework for evaluating the quality of included studies. This could include sample size justifications, statistical power analysis, or the use of pre-registration in animal studies. These elements can also introduce bias or minimize those if in place.

We agree these are important and the SYRCLE risk of bias tool we used addresses many major domains of bias mentioned by the reviewer (e.g., selection bias, performance bias, detection bias, attrition bias, reporting bias). For example, the SYRCLE item of “selective outcome reporting” domain address pre-registration by asking “Was the study protocol available and were all of the study’s pre-specified primary and secondary outcomes reported in the current manuscript?”. The SYRCLE Risk of Bias tool represents the current state of the art for risk of bias assessment in preclinical systematic reviews and aligns well with similar tools used clinically, such as the Cochrane Risk of Bias tool. Although the tool does not assess statistical power, we would note that this is considered to be a separate issue from internal validity, and it is the reason this is not even assessed by the Cochrane risk of bias tool used in clinical systematic reviews.

Please also highlight confounding factors that might have influenced the results in the included studies, such as variation in stroke models, dosing regimens, or behavioral assessment methods.

We agree that exploring potential confounding factors is an important element of the assessment. We highlight potential confounding factors in several parts of the Results and Discussion, such as in our Synthesis of Behavioural Outcomes section, “…equivalent infarct volumes were not demonstrated between the treated and control groups in this cohort, which could potentially lead to confounding effects.” and Comprehensiveness of Preclinical Evidence section, “All studies tested both behavioral and histological outcomes and demonstrated neuroprotective effects, but most studies failed to measure and control post-stroke temperature, which could potentially confound the observed neuroprotection (Table S4).(32) Most histological measurements were also assessed at <72 hours, which could confound the observed neuroprotective effects if cell death was merely delayed.(32) For CCR5 antagonists as a post-stroke recovery-inducing treatment, one experiment assessed the effects of initiating CCR5 administration in a similar post-stroke phase as the CAMAROS trial. This experiment (Joy et al.)(6) did not demonstrate that each treatment group had equivalent baseline stroke volumes, which may potentially confound observed behavioral effects.”

Although there are many factors that could potentially confound the observed results, we believe that we have addressed some of the most prominent examples that are known in the preclinical stroke literature. We expanded our statement in the final sentence of the Results to highlight this, “Overall, our assessments highlight a variety of knowledge gaps, potential confounding factors, and areas of misalignment between the preclinical evidence and clinical trial parameters that could be improved with further preclinical experimentation.

There is some discussion of the meta-analysis' limitations due to the few studies, but this point could be more thoroughly addressed. Please consider including a more critical discussion of the limitations of pooling data from heterogeneous study designs, stroke models, and outcome measures. What can this lead to? Is it reliable to do so, or does it lack scientific rigor? The authors are encouraged to formulate a balanced discussion adding, positive and negative aspects.

We appreciate the reviewer’s insightful comment regarding the limitations related to pooling data from heterogeneous study designs, stroke models, and outcome measures. We have added to the original limitations described in the first paragraph of our Discussion with additional text to provide a better balance about the potential risks and benefits of the meta-analysis strategy that we undertook in the present study.

“Pooling data across heterogenous experimental designs, animal/stroke models, and treatment parameters, as we have done with the infarct volume analysis in the present study, can introduce variability that increases the risk of overestimating or underestimating the true effect of the intervention.(38) Treatment effects observed across model systems and therapeutic compounds may represent different biological mechanisms. Despite this potential limitation, meta-analysis can provide valuable insights, especially in preclinical settings where the sample sizes of individual studies may be too small to detect significant effects on their own. In these cases, pooling data across studies can help identify overarching estimates of benefits and harm, highlight subgroups of interest, and help guide areas of future research. As described in the results above, we attempted to mitigate the risks of inappropriate data pooling through careful investigation of heterogeneity, subgroup analyses, and differentiation between outcomes where we felt that meta-analytic pooling was (infarct volume) and was not (behavioural outcomes) appropriate. Overall, we believe that our results indicate that further investigation is warranted to determine the optimal timing of administration and behavioral domains under which CCR5 antagonists exhibit the strongest post-stroke neuroprotective and recovery-inducing effects.”

The conclusion should more explicitly acknowledge that while CCR5 antagonists show potential, the findings are still preliminary due to the limitations in the preclinical studies (high bias risk, lack of diverse animal models). Overall, the conclusion can end with a call for rigorous, well-controlled, and replicated studies with improved alignment to clinical populations and trials to show that the conclusion remains inconclusive, considering what has been analyzed here.

We modified our concluding paragraph to highlight that the current evidence should be considered preliminary, as follows, “In conclusion, CCR5 antagonists show promise in preclinical studies for stroke neuroprotection, corresponding reduction in impairment, as well as improved functional recovery related to neural repair in the late sub-acute/early chronic phase. However, high risk of bias and the limited (or no) evidence in clinically relevant domains underscore the need for more rigorous and transparent preclinical research to further strengthen the current preliminary evidence available in the literature.”

**Reviewer #2 (Public review):**
Summary:This is an interesting, timely, and high-quality study on the potential neuroprotective capabilities of C-C chemokine receptor type 5 (CCR5) antagonists in ischemic stroke. The focus is on preclinical investigations.Strengths:The results are timely and interesting. An outstanding feature is that stroke patient representatives have directly participated in the work. Although this is often called for, it is hardly realized in research practice, so the work goes beyond established standards.The included studies were assessed regarding the therapeutic impact and their adherence to current quality assurance guidelines such as STAIR and SRRR, another important feature of this work. While overall results were promising, there were some shortcomings regarding guideline adherence.The paper is very well written and concise yet provides much highly useful information. It also has very good illustrations and extremely detailed and transparent supplements.Weaknesses:Although the paper is of very high quality, a couple of items that may require the authors' attention to increase the impact of this exciting work further. Specifically:Major aspects:(1) I hope I did not miss that (apologies if I did), but when exactly was the search conducted? Is it possible to screen the recent literature (maybe up to 12/2024) to see whether any additional studies were published?

We added the following statements to the “Information sources and search strategy” section of Materials and Methods to clarify the timing and intention of our search strategy, “The search was conducted October 25, 2022, to align with the listed launch date of the CAMAROS trial (September 15, 2022). Our intention in doing so was to collate and assess all preclinical evidence that could have feasibly informed the clinical trial. We sought to assess the comprehensiveness of evidence and readiness for translation of CCR5 antagonist drugs at the time of their actual translation into human clinical trials, as well as the alignment of the CAMAROS trial design to the existing preclinical evidence base.”

Although we agree that an update of the search provides valuable information for the field, we believe that the studies entering the literature after the launch of the CAMAROS trial fill a different conceptual niche than those prior to trial launch (since newer preclinical studies explicitly did not inform decisions to move to clinical trials or clinical trial design). It is our view that newer studies should be assessed from a lens of how effectively they close knowledge gaps that were present at trial launch and emulate the conditions of clinical trial populations and design parameters (which represent the de facto most “clinically relevant” conditions). Such an analysis would require a different approach that is outside the scope and aims of the present study. The present study provides an assessment of the preclinical literature up to the date of the translation of CCR5 antagonist drugs into human clinical trials (via the CAMAROS trial), which we believe will serve as a valuable prospective benchmark for evaluating the predictiveness of preclinical evidence after the results of the CAMAROS trial emerge.

(2) Please clearly define the difference between "study" and "experiment," as this is not entirely clear. Is an "experiment" a distinct investigation within a particular publication (=study) that can describe more than one such "experiment"? Thanks for clarifying.

We have now added definitions for “studies” and “experiments” immediately after the first time they are mentioned in paragraph one of the Study Selection section of Results, as follows: “Herein, “studies” refer to the published articles as a unit, while “experiments” refer to distinct investigations within each published article used to test various hypotheses (i.e., a subunit of “studies” comprised of a select cohort of animals).”

(3) Is there an opportunity to conduct a correlation analysis between the quality of a study (for instance, after transforming the ROB assessment into a kind of score) and reported effect sizes for particular experiments or studies? This might be highly interesting.

This is an interesting suggestion, which under different circumstances could provide insights into potential associations between study quality and effect size, as have been observed in the literature (e.g., Macleod et al., 2008; PMID:18635842). However, we are unable to assess this relationship in the present dataset as all studies were scored as “high risk of bias”, meaning that there was no variability in terms of observed study quality.

**Recommendations for the authors:**

**Reviewer #2 (Recommendations for the authors):**
Minor aspects:(1) The scope of the work is perfectly in line with very recent STAIR recommendations, which strongly suggest assessing potential interventions that may augment impact and improve outcomes in recanalization procedures (Wechsler et al., doi: 10.1161/STROKEAHA.123.044279; PMID 37886850). The authors may to discuss their work in light of these recent recommendations.

We thank the reviewer for highlighting the more recent STAIR recommendation document, as well as its focus on assessing interventions in conjunction with recanalization procedures. An item related to the importance of combining novel interventions with established recanalization procedures was included as part of Table S4 but was not highlighted in the main text. We have added to the final paragraph of the Results section “Comprehensiveness of preclinical evidence” to highlight that no studies tested CCR5 antagonist drugs in conjunction with recanalization procedures as follows, “…no studies assessed behavioural effects on upper extremity skilled reaching / grasping or potential interactions of CCR5 antagonists with rehabilitative therapies or established recanalization procedures (Table S4).(35–38)” The Weschler reference provided by the reviewer has now been cited as well.

(2) The authors may wish to consider the term "cerebroprotective" rather than "neuroprotective" unless neurons are the only cells to which a respective statement applies.

We agree that “cerebroprotective” is the more appropriate term and have thus substituted it wherever we previously used “neuroprotective”.

(3) The paper features a mixture between American (e.g.," hemorrhagic") and British English (e.g., "favours"). Although this is not untypical for Canadian English, deciding on one or the other may be an option.

Given *eLife*’s basis in the UK, we have modified the language used throughout to be consistent with British English style.